# Investigating the Design of a Photoplethysmography Device for Vital Sign Monitoring

**DOI:** 10.3390/s25061875

**Published:** 2025-03-18

**Authors:** Anneri Appel, Rensu P. Theart

**Affiliations:** 1Institute of Biomedical Engineering, Stellenbosch University, Matieland 7600, South Africa; anneri.appel@gmail.com; 2Department of Electrical and Electronic Engineering, Stellenbosch University, Matieland 7600, South Africa

**Keywords:** photoplethysmography (PPG), PPG sensor, optical sensor, wearables, signal processing

## Abstract

There exists a distinct lack of publicly available literature addressing the most effective hardware design for photoplethysmography (PPG) devices for clinical and domestic applications. In this article, this problem was addressed by investigating the hardware design configuration of a PPG device, with particular emphasis on the light source wavelength, light source brightness, number of light sources, photodetector lens shape, and sensor-to-skin contact pressure. A participant study was conducted to collect cardiovascular metric data from 110 participants with varying skin tones, which was used to determine the most promising hardware configuration of the PPG device. It was concluded that the device had little bias to skin tone, with only a 3.82 dB variance over all the skin tones tested.

## 1. Introduction

The Global Burden of Disease Study reports that cardiovascular disease has remained the leading cause of global mortality over the past two decades [1]. In South Africa, cardiovascular disease was responsible for 18% of all deaths in 2018 [2] and was reported as the leading cause of death in the same year [3]. Active monitoring of cardiovascular parameters using technologies such as photoplethysmography (PPG) can play a crucial role in addressing this pressing health concern.

PPG is a non-invasive optical technique that is used to detect changes in blood volume in the microvascular bed of tissue [4]. This is achieved by measuring the attenuation of light energy through the tissue in the vicinity of the PPG device [5], which primarily consists of a light source and a photodetector. Transmissive devices, like pulse oximeters, are designed with the photodetector and light source placed on opposite sides of the tissue, and therefore can only be used on the user’s extremities. However, reflective devices, like the PPG sensors of smartwatches, offer greater site versatility due to placing the photodetector and light source on the same side of the tissue [4].

PPG data are already widely used in both clinical and domestic devices to measure heart rate, blood oxygen saturation, and other cardiovascular metrics [6,7,8]. However, the quality of the data influences the accuracy of these estimations and therefore determines its usefulness in managing cardiovascular health [9]. One such factor of concern is the influence of skin tones on PPG signal quality, which was highlighted by problems experienced by dark-skinned patients during the COVID-19 pandemic [10,11,12]. This calls for improved designs to minimise biases and optimise hardware configurations for accurate cardiovascular monitoring [13,14].

Drawing on insights from two key studies, this article incorporates the findings that guided its design choices: The first study validated the investigation of light source wavelengths on PPG data quality and recommended an LED-to-photodetector distance of 6 mm [15]. The second provided evidence that higher contact pressures can enhance PPG signal quality [16].

The primary aim of this article is to explore various PPG hardware designs to develop a wearable cardiovascular monitor with a refined hardware configuration. By investigating variables like light source wavelength, photodetector lens shape, and contact pressure, the article seeks to enhance the device’s performance for various skin tones. The effectiveness of these hardware configurations was assessed through a preliminary performance evaluation and participant study.

## 2. Hardware Design

A PPG device was designed to allow for variations in hardware configurations, enabling an investigation of performance characteristics during the preliminary evaluation and participant study. The hardware design was divided into three sections: the circuitry design; the printed circuit board (PCB) design; and the enclosure design. The outcomes of each of these sections are herein described in detail.

### 2.1. Circuitry Design

The input circuit of the PPG device refers to the circuitry designed to capture the photodetector’s signal, amplify it, filter it, and send it to the analogue-to-digital converter (ADC) to be captured digitally on the ESP32-Pico-V3 [15]. This circuit was divided into three separate circuits: the transimpedance amplifier (TIA); the low-pass filter; and the high-pass filter. It is important to note that the output from the TIA is not directly fed into the ADC. Instead, it undergoes further amplification in the low-pass filter stage, ensuring that the final signal level is suitable for digitization. Since different responses were required of the high- and low-pass filters, they were designed separately instead of implementing a band-pass filter. Each filter was designed using the Filter Design Tool provided by Texas Instruments [16] and analysed using pole-zero and root locus analyses, which are provided in Appendix B. The final versions of these sub-circuits are illustrated in Appendix A. The final circuit design of the complete circuit is illustrated in Appendix A.

A transimpedance amplifier (TIA) is a circuit that converts a current input to a proportional output voltage. The typical components of a TIA are a photodetector, feedback resistor, feedback capacitor, and a central operational amplifier (op-amp).

For the prototype developed in this study, Onsemi’s QSB34CGR photodetector was chosen for its wide spectral sensitivity (400 to 1100 nm) and small footprint (3 × 3 mm) [17]. The output current of the photodetector is rated between 0 and 37 µA. Furthermore, Texas Instruments’ OPA170 op-amp was selected due to its 2.7 to 36 V supply range, low noise, rail-to-rail output, and low-bias current [18]. The low-bias current allows the accurate measurement of small current and voltage signals, which is essential for capturing the current signal from the photodetector in this application [19].

During testing, it was observed that signals closer to 0 V were clipped. Although the OPA170 does have rail-to-rail operation [20], in practice, it is very common for rail-to-rail op-amps to clip signals that are close to the rail value. This meant that data were lost when the current signal was converted to a voltage signal through the TIA. To contend with this, a small-bias voltage of 4 mV was added to the standard TIA’s design.

Additionally, the original design included a feedback capacitor with a high-frequency cut-off of 10 Hz (equivalent to the highest recorded heart rate, 600 BPM [21]). Testing determined that this feedback capacitor caused instabilities in the TIA’s performance, regardless of its stable theoretical model. Therefore, the feedback capacitor was removed, and its amplification functionality was compensated for in the design of the low-pass filter. Other nuances that could be related to the removal of the capacitor were not observed during testing. Furthermore, the amplification of the TIA was lowered from 100,000 V/V to 100 V/V, as lower amplification provided better performance, and additional amplifications would be completed by the low- and high-pass filters that follow the TIA.

The design of the low-pass filter that followed the TIA was that of a fourth-order Butterworth Sallen–Key filter. It was originally designed with a cut-off frequency of 10 Hz and an amplification value of 100 V/V but was reduced to 10 V/V for further stability concerns. This 10 V/V amplification stage not only stabilizes the signal but also boosts the TIA output to a level that is well suited for subsequent digitization by the ADC.

Finally, the high-pass filter design was that of a second-order Butterworth Sallen–Key filter, with a cut-off frequency of 0.5 Hz (roughly equivalent to the lowest recorded heart rate, 27 BPM [22]). The lower order of this filter was selected due to the minimal desire to attenuate the DC components of the PPGs. Originally, it had an amplification value of 100 V/V, but it was observed that the performance of the filter was improved at 50 V/V, considering the changes made to the TIA and low-pass filter during testing.

The output signal of the final input circuit design was recorded by an oscilloscope, as illustrated in Figure 1. This was measured by placing an index finger against the photodetector and illuminating the finger with a smartphone LED opposite the photodetector, therefore on the outside of the finger.

### 2.2. PCB Design

With the design of the PCB, it was first important to take note of the considerations related to the application of the PCB in the article. To ensure proper operation, the PPG device’s PCB must press the LEDs and photodetector against the user’s skin. Human skin is rated normally between 33 and 37 °C [23]. Thus, to prevent more complex components like the ESP32 chip from overheating, the PCB was designed to be double-sided. This ensured that the LEDs and photodetector could be pressed against the user’s skin while keeping more sensitive components on the other side of the PCB. This design decision also decreased the overall size of the PCB.

The PCB design involved various sections, including the following:The voltage-regulator circuit and accompanying connections;The microcontroller and its flash-circuit, where a flash-circuit allows the developer to program the microcontroller;The input circuit;The LED circuit;The ADC circuit;The antenna circuit.

Because each circuit has its function, the components were positioned in such a way that each circuit’s respective components were positioned close together while keeping the individual circuits separate. Furthermore, the microcontroller circuit was positioned in the centre of the “top side” of the PCB for better heat distribution and ventilation. For clarification, the “bottom side” of the PCB is the side that makes direct contact with the user’s skin.

The design of the PCB should allow for the smallest footprint practically obtainable. For this reason, a four-layer PCB design with internal 3.3 V and ground layers was implemented. From previously conducted research, it was suggested that a circular LED circuit should be positioned with a 6 mm gap around a central photodetector [24]. In this study, the circular LED circuit included eight LEDs paired diagonally according to colour (infrared and either red or green). The LEDs used were SunLED’s surface-mount 2.0 × 1.25 mm infrared [25], green [26], and red LEDs [27]. These wavelengths are further explained in Section 4.1.1.

The input circuit discussed in the previous section described the circuit design that connected the photodetector to the input of the ADC. This included a TIA with an OPA170 op-amp, followed by two filter circuits consisting of a total of three op-amps, which were designed using the MCP6024 op-amp [28]. The number of components initially promoted the idea of placing these circuits on the “top” of the PCB and using a via to connect the photodetector to the TIA, as it would also allow for better ventilation for the op-amps. However, since the signal that was outputted from the photodetector was not only very small, but its integrity was also essential for the operation of the device, the trace between the photodetector–output and TIA–input had to be kept as short as possible to prevent the induction of noise onto the signal. To achieve this, the TIA circuit was placed next to the photodetector on the “bottom” of the PCB, with a via connection to the filter circuit on the “top”.

During the testing of the manufactured PCB, the quality of the signals measured by the ESP32’s 12-bit resolution ADC was investigated. As suspected, the ESP32’s ADC was not sufficient for this application of signal capturing and reported only nonsensical data. Therefore, it was deemed necessary to implement an external ADC, the ADS1115 with a 16-bit resolution [29].

The final circuit with a specific placement requirement was that of the antenna, KH5220-A36 [30]. This circuit required a keep-out area, which meant that no components or traces could infringe on that specific area on any of the PCB’s layers. Although this was impossible due to the size of the design, every attempt was made to minimize the number of components and traces that breached the outer edges of the area. The worst instance of this infringement was the trace of an input–output pin running directly below the ground pin of the antenna, on the “bottom” of the PCB. This breach was only allowed due to the short distance that the antenna had to broadcast. If the antenna had to broadcast over larger distances, which might be the case in future work, any breaching of the keep-out area could have impacted the operation of the antenna more severely and would have had to be removed entirely.

In addition, the trace from the antenna to the applicable microcontroller pin also had to be as short as possible to reduce the amount of interference on the line. Finally, the PCB had to be designed so that no components, like batteries, covered the antenna.

The final design of the PCB accounted for all the considerations mentioned in this section and is provided in Figure 2 and Figure 3.

### 2.3. Enclosure Design

The enclosure design used in this study was for a system designed for wired operation, although a wireless battery-operated enclosure design was also developed. The reason the battery-operated design was negated was to allow the data to be transferred to an external computer via both wireless and wired communication by using a file transfer protocol (FTP) server and universal asynchronous receiver/transmitter (UART) serial communication. As an additional note, each enclosure design had to account for proper ventilation, user comfort, and a method of attaching to the user’s body.

Both enclosures had a simplified smartwatch design, which was modified to fit the component assemblies and adhere to the requirements stated above. Like a smartwatch, the enclosures had strap-fits designed to the upper half of the enclosure, allowing nylon straps to attach the device to the user’s body. To provide proper ventilation, the upper halves of the enclosures were designed with ventilation holes on the roof and the sides, with a larger cutout on the wired design to allow wires to exit the enclosure. The holes on the sides were designed to fit a standard lever switch to switch the device on or off if a battery was connected. Finally, user comfort was achieved by chamfering any sharp edges and keeping the contact area flat.

Additionally, the lower half of the enclosure had cut-outs to allow the LEDs and photodetector to contact the user’s skin but was otherwise completely enclosed. The two designs of the enclosure can be seen in Figure 4, with detailed technical drawings in Appendix A.

## 3. Software Design

This section provides a brief overview of the software designed for this study. All the relevant software for this article is available at https://github.com/FridgeMagnetPoet/Investigating-the-Design-of-a-Photoplethysmography-Device-for-Clinical-Applications.git (accessed 13 March 2025).

### 3.1. Data Acquisition Software

The data acquisition software refers to the software implemented on the ESP32 to capture the PPG data. The primary language and integrated development environment (IDE) for this software was C++11 and the Arduino IDE, version 2.2.1. The software was flashed onto the ESP32 using the PCB flash-circuit header designed into the hardware. There were two stages wherein data acquisition was implemented: the preliminary performance evaluation and the participant study.

The data acquisition software had three main functions. The first was to initialise and set up all the functionalities utilised by the ESP32 during operation. The second involved turning the LEDs on and off as required by the objectives of each stage of the study while also digitally measuring the intensity of the reflected light with the ADC. The final function was sending the data to an external device for processing and storing.

### 3.2. Data Processing Software

The data captured during the preliminary performance evaluation and the participant study was processed using Python 3.9 scripts in Visual Studio Code, version 1.85. In each instance of processing software, the script extracted the data from the text files and trimmed the data due to the unstable transient response of the system during the first couple of seconds of measurement (see Figure 5 for a graphical illustration of this phenomenon) and subtracted the ADC values from 65,535 to invert the signal as a unit of light absorption.

### 3.3. Cardiovascular Metric Estimation Software

Following the conclusion of the study, the most promising PPG device could be used to estimate various cardiovascular metrics, notably heart rate, blood oxygen saturation, and blood pressure. For blood pressure estimation, a neural network would have to be trained. Here, it is strongly suggested that arterial blood pressure be used as a golden standard, as blood pressure cuff measurements are too inaccurate to train models on.

## 4. Preliminary Performance Evaluation

In this section, the preliminary performance evaluation of the different PPG device configurations is described. This evaluation’s primary goal was to eliminate hardware configurations based on their performance to ensure quality data collection during the participant study. This section discusses the various device configuration categories and insights acquired. Further information regarding the particulars of the study can be found in Appendix C.

### 4.1. Device Configuration Categories

A total of four variations of the PPG device with different hardware configurations were tested in the preliminary performance evaluation. The following is a list of the different hardware characteristics as well as their variations. These characteristics are explained in the forthcoming subsections.

It is important to note that the selection of these configuration categories and their variations was not arbitrary but rather informed by established principles in the PPG device design literature. Each category represents key design parameters that prior research has shown to significantly impact PPG signal quality. The specific variations within each category were selected based on three primary considerations: (1) documented performance differences in previous studies, (2) practical constraints of commercially available components suitable for wearable devices, and (3) the need to represent a meaningful range of options within reasonable testing limits. By systematically evaluating these carefully selected parameters, we aimed to identify optimal configurations for different use cases while maintaining scientific rigor in our approach.

Light source wavelength: Red, green, or infrared;Light source brightness (8-bit duty cycle values): 100, 175, or 250;Number of light sources: One pair or two pairs;Photodetector lens shape: Flat or dome;Sensor-to-skin contact pressure: Low, medium, or high;Body location: Finger or wrist.

#### 4.1.1. Light Source Wavelength

Human skin consists of chromophores, which scatter and absorb light depending on the wavelength. This results in different wavelengths of light penetrating the skin to different depths [31], with increasing wavelength resulting in deeper penetration [32]. Additionally, melanin plays a key role in light absorption, increasing with shorter wavelengths [33]. Therefore, this study tested the influence of light source wavelength on the quality of PPGs measured across different skin tones. Three wavelengths were selected for testing, namely green (515 nm), red (640 nm), and near-infrared (940 nm), which is generally referred to as “infrared” in the text. As both infrared and a visible-spectrum light source are required for blood oxygen saturation estimations, the evaluation aimed to identify the best-performing visible-spectrum LED to pair with infrared.

#### 4.1.2. Light Source Brightness

To determine the effects of the light source brightness on the quality of the PPG data captured, three brightnesses were tested in terms of pulse width modulation (PWM) duty cycle. These brightnesses were empirically chosen to be the 8-bit values of 100, 175, and 250 out of 255 due to the significant observable difference in brightness of each during testing. According to a previous study, decreasing infrared light intensity should result in noisier PPGs [34]. This supported the hypothesis that the LEDs’ brightness could influence the quality of the PPG signal, which was tested for the red, green, and infrared LEDs, respectively.

#### 4.1.3. Number of Light Sources

The PPG device was designed with eight LEDs—four infrared and four visible-spectrum LEDs (either red or green). To determine the influence of the amount of incident light projected into the tissue on the quality of the PPGs measured, two arrangements were tested. In the first arrangement, one pair of active LEDs was used, meaning that two diagonal infrared and two diagonal visible-spectrum LEDs were activated for their respective measurement periods. In the second arrangement, two pairs were activated, which meant that all four infrared and visible-spectrum LEDs were active, respectively.

It was hypothesized that increasing the incident light might improve the capture of high-quality PPG signals from the surrounding blood vessels, particularly in non-extremity areas. However, it was also considered that an excessive amount of light could increase scattering, potentially degrading the signal quality. This practical investigation was therefore necessary to determine the true effects of the number of light sources on PPG quality.

#### 4.1.4. Photodetector Lens Shape

Two different photodetectors were tested to determine how the shape of the lens influenced the quality of the PPG data. The first photodetector has a flat lens [17], while the second has a dome-shaped lens [35]. Each photodetector had a wide spectral sensitivity range to ensure that the green, red, and infrared wavelengths could all be accurately captured so as not to influence the data.

#### 4.1.5. Sensor-to-Skin Contact Pressure

A previous study described the effect of contact pressure on the quality of PPG measurements. In that study, the researchers examined three different contact pressures using a loadcell: 12, 33, and 54 mmHg [36]. Since there was not a loadcell built into the design of this study’s PPG device, the contact pressure could not be controlled quantitatively. Instead, the following description for contact pressure when attempting to lift the fastened PPG device from the user’s skin was used:Low pressure: The device can easily lose contact with the user’s skin;Medium pressure: The device stays flush with the user’s skin but can be moved from side to side;High pressure: The device does not lose contact with the user’s skin and cannot easily be moved from side to side.

A previous research article concluded that the highest contact pressure tested (54 mmHg) provided the most accurate PPG results. It was therefore also hypothesised that this study would have similar results. Note that beyond a certain pressure threshold, the arteries will start to compress and constrict blood flow [37], thereby decreasing the quality of the measured PPG signal.

#### 4.1.6. Body Location

Since one aim of the device was for it to be capable of use on various locations of the body, the performance of the configuration variations on different locations had to be tested. For simplicity and comfort, the locations chosen for this experiment were the insides of the wrists and the index fingertips.

### 4.2. Ethical Approval

Due to the data collection of the study requiring the involvement of participants, it was necessary to obtain ethical approval from the Health Research and Ethics Committee of Stellenbosch University. The Project ID was 27,668 and the Ethics Reference Number S23/06/142. As the preliminary study was closed, and no consent was given for data sharing, the data will not be made available with this article. For the participant study, however, consent was given to share the data publicly.

### 4.3. Participants

Nine participants were recruited for this experiment, most with varying skin tones according to the Monk Skin Tone scale. The nine participants had skin tones ranging from 2 to 9, with two participants rating at 6.

Note that participants were recruited on a voluntary basis. Therefore, the evaluation cannot comment on the influence of abnormal anatomical or physiological cases on the quality of PPGs measured, as it was beyond the scope of the research.

### 4.4. Evaluation Insights

Only six participants’ data were used for this evaluation since the other three datasets were eliminated due to excessive movement. These six participants all had different skin tones according to the Monk Skin Tone scale. The evaluation process followed a systematic approach to identify optimal device configurations through both statistical and qualitative analyses. Statistical significance was assessed using analysis of variance (ANOVA) and *t*-tests, with a significance level of α = 0.05. While these tests often yielded no statistically significant differences between hardware configurations, we complemented this with qualitative analysis of signal characteristics to make informed design decisions.

Specifically, our evaluation methodology employed a hierarchical elimination approach whereby each configuration category was considered, such as body location, light source wavelength, number of light sources, light source brightness, photodetector lens shape, and sensor-to-skin contact pressure, and the less effective options were eliminated before proceeding to the next category. In cases where no clear winner emerged, all options within that category were retained for further evaluation. For instance, Figure A10 presents the average values of the FFT and time signals’ quality factors for each hardware configuration, arranged in increasing order of participant Monk Skin Tone, covering categories including (a) light source wavelength, (b) number of light sources, (c) photodetector lens shape, (d) sensor-to-skin contact pressure, (e) light source brightness, and (f) body location. Our quality factor was a composite metric derived from both time-domain and frequency-domain characteristics of the PPG signals, where lower values indicate better signal quality. Furthermore, to support our qualitative analysis and enhance objectivity, Figure A8 and Figure A10 in Appendix C illustrate representative signals exhibiting different evaluation quality factors, clearly demonstrating the criteria used for qualitative assessment.

The following subsections describe the insights for each of the device configuration categories. Note that the examples in Figure 6, Figure 7, Figure 8, Figure 9, Figure 10, Figure 11, Figure 12 and Figure 13 are included to provide a concise overview of the outcomes and are not the sole basis for the conclusions drawn.

#### 4.4.1. Body Location

It was obvious from the investigation that the index fingertip provided better-quality PPG signals than the wrist, as seen in the PPG signals illustrated in Figure 6. This was expected due to two reasons. First, the index finger is an extremity with good blood flow and relatively thin skin in the fingertip area. There are also not any large muscles in the fingertip which could induce motion artefacts into the PPG data. The wrist, which specifically refers to the anterior proximal area of the wrist joint, only has two major arteries: the radial and ulnar arteries. Both are semi-obscured by muscles, which induce motion artefacts.

In addition, the two-sample *t*-test reported a statistically significant difference between the means of the data captured by the fingertip and that captured by the wrist (*p* = 0.003121). Due to this, the devices only measured the index fingertip for the subsequent data acquisition in the participant study.Figure 6Body location example PPG signals. Quality factors were calculated for each signal and used in statistical comparisons between body locations.
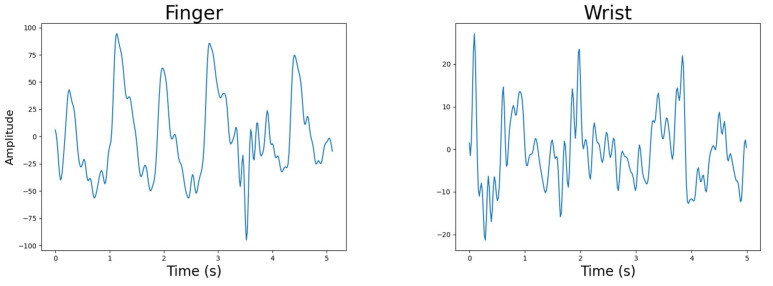


#### 4.4.2. Light Source Wavelength

By evaluating the light source wavelength data measured at the index fingertip, it was concluded that the distinction between the red and green visible-spectrum LEDs’ performance was not sufficient to validate selecting one over the other (*p*_ANOVA_ = 0.604206 for the entire dataset). The infrared LED, however, performed better than both the visible-spectrum LEDs on a qualitative basis. This is illustrated by two clear examples in Figure 7 and Figure 8. Note that the device, participant, body location, number of light sources, light source brightness, and sensor-to-skin contact pressure were the same for both signals in Figure 7 and similar for the two signals in Figure 8. However, the instance of time differs between paired signals in each figure.

As a note, to explore the reason behind the acceptance of the null hypothesis for the *t*-tests in this category, further consideration was then given to validate our statistical approach. We conducted power analyses for each comparison. Here, the power of red vs. green was 0.88, infrared vs. green was 1, and infrared vs. red was 1. Given the high statistical power of the *t*-tests in this category, the sample size was deemed sufficient for detecting any true effects. The acceptance of the null hypothesis suggests no significant differences between the groups under study. Consequently, further statistical analysis may not provide additional insights into the data. Qualitative analysis was therefore used to identify subtle signal quality differences not captured by the statistical tests.Figure 7Red LED versus infrared LED example PPG signals. Quality factors were calculated for each signal and used in statistical comparisons between LED wavelengths.
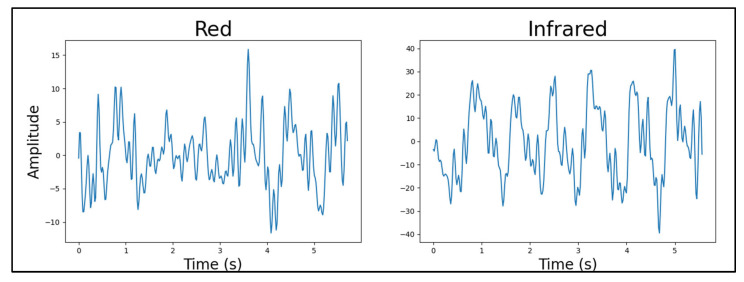

Figure 8Green LED versus infrared LED example PPG signals. Quality factors were calculated for each signal and used in statistical comparisons between LED wavelengths.
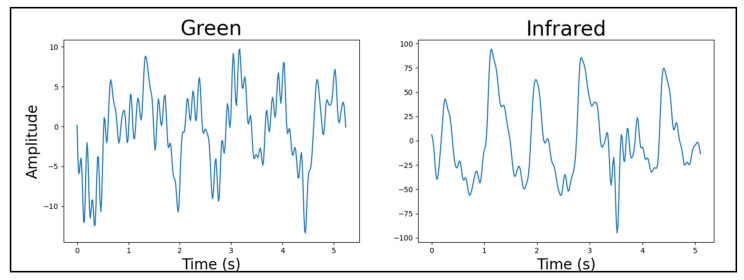


The outperformance of the infrared LEDs was expected since their penetration depth into the tissue was much larger than the visible-spectrum LEDs’. This prompted the idea of future devices only using infrared LEDs for heart rate estimation, as the PPG signals were of the best quality. However, to estimate blood oxygen saturation, a visible-spectrum LED had to be present as well. Therefore, the visible-spectrum LEDs were separated into two device configurations: one with red and the other with green LEDs, each with a set of infrared LEDs.

#### 4.4.3. Number of Light Sources

Since both the visible-spectrum and infrared LEDs were still under evaluation, each wavelength was separately evaluated for the best-performing number of light sources. For both the visible-spectrum LEDs, the data captured by two pairs of light sources (four LEDs) were slightly better than the data for one pair. For the infrared LEDs, however, one pair performed better than two. This could not be proven statistically, as the null hypothesis of the *t*-test was not rejected (*p* = 0.909). This result was likely due to an insufficient sample size; the test would have required a sample size of 3440 instead of 864 to increase the statistical power from 0.29 to 0.8.

It was not possible to provide a definitive reason as to why the visible-spectrum LEDs performed better with more incident light than the infrared LEDs. However, it was suspected that because the visible-spectrum LEDs have lower wavelengths and therefore shorter penetration depths, more incident light did indeed increase the likelihood of high-quality PPGs being measured in surrounding blood vessels. In the case of the infrared LEDs, however, more incident light at deeper penetration depths possibly caused more biological artefacts to be introduced into the data.

#### 4.4.4. Light Source Brightness

For the red LEDs, the best-performing light source brightness was at a duty cycle of 175 (Figure 9). For green, it was at the lowest value of 100 (Figure 10) and for infrared at the highest value of 250 (Figure 11). The result of the infrared’s brightness correlated with the previous research’s observation [34], which stated that the intensity of the infrared light was concurrent with the quality of the PPG data.

Note that it was also observed that the brightness did not affect the quality of the green LEDs as significantly as it did red and infrared, and with increasing penetration, the highest quality PPG was captured with increasing brightness. One possible reason for this outcome is the scattering of each wavelength. This wavelength-dependent behaviour can be explained by light-scattering and penetration-depth principles. Since the tissue penetration depth of light increases with increasing wavelength, lower penetrative wavelengths would scatter more superficially. Thus, it is logical to assume that, for example, infrared would provide higher-quality PPG signals at a higher brightness, as the excess scattering happens further away from the photodetector. Similarly, red, which has a shorter wavelength than infrared and a longer wavelength than green, would provide better results at a brightness level lower than infrared and higher than green, as the scattering occurs at a penetration depth between that of green and infrared.

Similar to the light source wavelength test, the failure in rejecting the null hypothesis in the ANOVA and *t*-tests provided no statistical evidence of a difference between the datasets (*p*_ANOVA_ = 0.87584). For these brightness comparisons (Figure 9, Figure 10 and Figure 11), we conducted pairwise *t*-tests comparing the means of our PPG signal quality factors across three brightness levels (duty cycles of 100, 175, and 250) for each LED type (red, green, and infrared). Specifically, we performed three pairwise comparisons for each LED type: 100 vs. 175, 175 vs. 250, and 100 vs. 250. Our quality factor was a composite metric derived from both time-domain and frequency-domain characteristics of the PPG signals, where lower values indicate better signal quality with fewer artifacts and clearer physiological features (detailed calculation methodology available in Appendix C). Our power analysis showed that in the case of 100 vs. 175 and 250 vs. 175, a sufficient number of samples was provided for the test, with powers larger than 0.9. However, in the case of 100 vs. 250, a sample size of 913 instead of 864 would be required to raise the power from 0.77 to 0.8. Despite the lack of statistical significance, the qualitative differences observed in signal quality informed our final device configurations.Figure 9Red LED brightness example PPG signals, from 100 to 250 brightness. Quality factors were calculated for each signal and used in statistical comparisons between brightness levels.
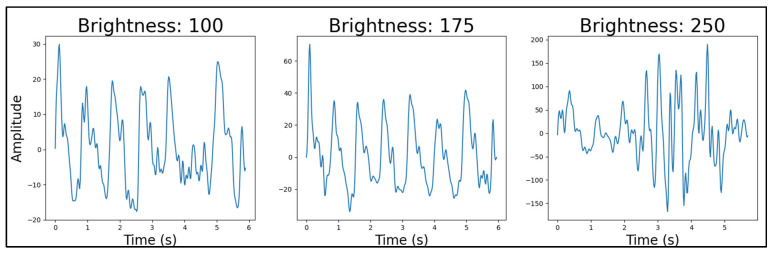

Figure 10Green LED brightness example PPG signals, from 100 to 250 brightness. Quality factors were calculated for each signal and used in statistical comparisons between brightness levels.
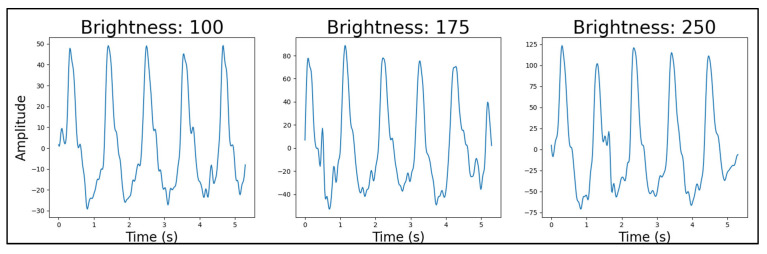

Figure 11Infrared LED brightness example PPG signals, from 100 to 250 brightness. Quality factors were calculated for each signal and used in statistical comparisons between brightness levels.
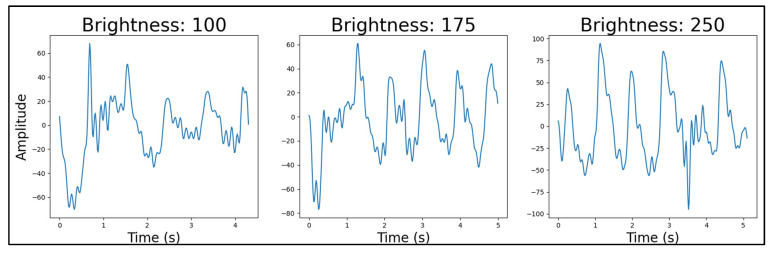


#### 4.4.5. Photodetector Lens Shape

Both the red–infrared and green–infrared PPG devices captured the best-quality PPG data with a flat lens photodetector as opposed to the dome-shaped lens, as illustrated in Figure 12. This was evident from the quality analysis as well as the statistical significance provided by the *t*-test (*p* = 0.035749). Again, a definitive reason for this observation could not be made, but it was suspected that with an increasing collection of light at different angles, the dome-shaped lens’s data were influenced by biological and environmental noise. A secondary observation was that the participants experienced discomfort from the dome-shaped lens due to it pressing deeper into the skin, and therefore, it would not be recommended for future PPG devices intended to be worn for extended periods.Figure 12Photodetector lens shape example PPG signals.
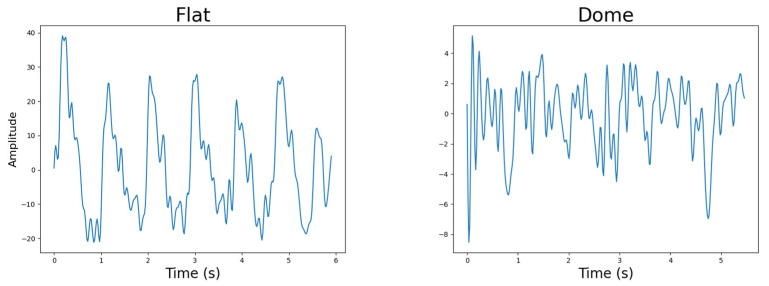


#### 4.4.6. Sensor-to-Skin Contact Pressure

The final category evaluated for the two separate devices was contact pressure. As expected from the results of the previously discussed study, the highest contact pressure provided the best overall PPG data for both the red–infrared and green–infrared PPG devices. It is worth noting that the quality of the data did increase with increasing contact pressure, as was also shown by a previous work [36]. Therefore, the low contact pressure performed the worst, while the high contact pressure performed the best. This is also illustrated in Figure 13.

This, however, was not evident from the statistical tests performed, as the null hypothesis was not rejected in any of the comparisons. For the low vs. medium, low vs. high, and medium vs. high tests, the statistical power was 1 in each case. Therefore, the sample size was sufficient for the tests to draw reliable statistical conclusions. The discrepancy between qualitative observations and statistical results suggests that while trends are visually apparent, the variability in the data prevented statistical significance, highlighting the importance of combining both approaches in PPG device evaluation.Figure 13Sensor-to-skin contact pressure example PPG signals for low to high pressure.
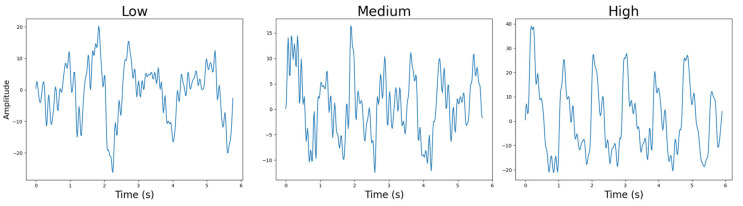


#### 4.4.7. Skin Tone

In this evaluation, some care was taken to also investigate the influence of skin tones on the different categories. However, this was inconclusive due to the small size of the participant group. It appeared from the evaluation insights that the performances of the different categories increased to the middle of the Monk Skin Tone scale, with the noisier data belonging to participants with lighter and darker skin tones. Yet, it cannot be excluded that factors such as age, sex, or even inconsistencies in the measurement process influenced this observation. Given this limitation in our preliminary evaluation, a more robust investigation with a larger, diverse participant pool was necessary. As a result, this was further investigated in the participant study, where a much larger group of participants were evaluated.

### 4.5. Evaluation Conclusion

In this preliminary performance evaluation, it was determined that there was not one device configuration that performed the best. Rather, two device configurations were identified and tested in the ensuing participant study, whose configurations are summarised in Table 1. Furthermore, the performance of the devices across different skin tones was inconclusive and was also further investigated in the participant study.

## 5. Participant Study Results

This study evaluated the quality of PPG signals across various skin tones using the two device configurations outlined in Table 1. This section presents and analyses the results of the participant study, identifying the best-performing PPG configuration. Additionally, it examines the device’s bias and performance variations across different skin tones and body locations. Further details on the study methodology can be found in Appendix D.

In total, 110 participants took part in the participant study. Participants’ skin tones, according to the Monk Skin Tone scale, ranged from 1 to 8.

Note that participants were recruited on a voluntary basis. Therefore, the study cannot comment on the influence of abnormal anatomical or physiological cases on the quality of PPGs measured, as it was beyond the scope of the research.

### 5.1. PPG Device Configuration

There were two metrics used to determine the best-performing PPG device configuration. These metrics included Pearson correlation coefficient and signal-to-noise ratio (SNR). To properly calculate these metrics, it was first important to group the processed signals of the two devices that were measured simultaneously. The first group of signals comprised the red LED PPG data measured by the first device and infrared LED PPG data measured by the second device, respectively. In turn, the second group of signals included the green LED PPG data measured by the second device and infrared LED PPG data measured by the first device, respectively.

As already observed in Section 4.4.2, the infrared LED performed much better than the visible-spectrum LEDs. In other words, it had the cleanest PPG data. Therefore, the infrared signals were selected to act as the reference signals against which the visible-spectrum data would be evaluated. This was deemed appropriate since even though the visible-spectrum and infrared data that were grouped were measured by PPG devices with different configurations, the configurations of the infrared measurements were the same.

The metric used to investigate the relationship of these signals was the Pearson correlation coefficient, which quantifies the strength of the linear association of the two signals. Stronger, positive linear relationships will tend to a coefficient of 1, while the lack of a linear relationship will tend to 0. Should a strong, negative linear relationship exist, however, the coefficient will tend to −1 [38]. In this evaluation, the signal from the red or green LED device configurations with a higher Pearson correlation coefficient would suggest its superior performance.

The results of this metric’s investigation for each of the device configurations are given in Table 2.

Based on these results, both the red LED and green LED PPG device configurations’ visible-spectrum signals had strong, positive relationships with their infrared reference signals. However, the red LED PPG device outperformed the green, which was expected. This is because the red light wavelength penetrates the tissue deeper than the green.

To further this evaluation, the SNR (signal-to-noise ratio) of each different device configuration was determined. The SNR is the measurement of the strength of the signal relative to the noise; therefore, higher SNR values indicate a better signal quality [39]. In this application, the SNR was calculated as follows:(1)$$SNR=20\log_{10}\frac{P_{S}}{P_{N}}$$
where *P_S_* is the power of the reference signal (infrared PPG), and *P_N_* is the power of the noise signal. The noise signal was determined as follows:(2)Noise=Reference Signal−Test Signal
where the *Reference Signal* is the infrared LED’s PPG signal, and the *Test Signal* is the visible-spectrum LED’s PPG signal. The power of the signals was calculated as follows:(3)$$P=\frac{1}{N}\sum_{i=1}^{N}{x_{i}}^{2}\;$$
where *N* is the total number of datapoints, and *x* is the signal.

The results of the average SNR calculation for each device configuration are given in Table 3.

The average SNR values for the two device configurations also showed that the red LED PPG device produced better-quality PPG signals compared to the green LED device. Therefore, the red LED PPG device configuration was selected as the best-performing PPG device configuration in this study.

### 5.2. Bias to Skin Tone

The average estimated signal-to-noise ratio (SNR) of the data was once again used to determine the quality of the PPG data over different skin tones. Again, this SNR was estimated by using the infrared PPG signal as a reference and the red PPG as the test signal. As a result, the average estimated SNR values are reported in Table 4 and illustrated in Figure 14.

Specifically focusing on the trendline of the diagram, it is evident that there was a slight increase in PPG quality towards darker skin tones. Given that, historically, darker skin tones have been subject to a negative PPG measurement bias, this outcome was unexpected [40]. However, there was only a 3.82 dB difference between the average estimated SNR values of the various datasets and Monk Skin Tone variations. Consequently, the bias that the device has towards skin tone was deemed sufficiently low.

Note that in the calculation of these estimated SNR values, both the reference and test signals were subject to error. Therefore, the reported bias is subject to debate, and it is suspected that a different group of participants could yield a different outcome. Future research should focus on measuring vital signs with gold-standard devices and comparing those values with the ones estimated from PPG signals.

## 6. Conclusions

This study underscores the importance of rigorous hardware design evaluation in developing accurate and inclusive photoplethysmography (PPG) devices for clinical applications. By systematically investigating key hardware parameters—including light source wavelength, brightness, number of sources, photodetector lens shape, and sensor-to-skin contact pressure—we identified an optimized PPG device configuration that performs reliably across diverse skin tones.

The study followed a two-stage evaluation process: an initial preliminary performance assessment to eliminate underperforming configurations, followed by a participant study with 110 individuals. The findings indicate that a red–infrared LED PPG device provided the most consistent and accurate vital sign monitoring across varying skin tones. Interestingly, while the device exhibited minimal bias to skin tone, a slight improvement in performance was observed in participants with darker skin tones.

These findings contribute to the advancement of PPG hardware optimization, highlighting the need for careful design considerations to enhance accuracy, reduce bias, and ensure more inclusive clinical applications.

### 6.1. Article Contributions

The contributions of this article were the following:A promising hardware configuration of a PPG device: This includes both the electronic circuit and printed circuit board (PCB) design, with full schematics published with this work;A dataset of the heart rate, blood oxygen saturation, blood pressure, and PPG data of 110 participants.

These contributions can be used as a foundation for further research to explore new applications of PPG devices for clinical applications.

### 6.2. Future Work

There are various suggestions for future work that stem from the conclusion of this article. The first set of suggestions is specifically in terms of the hardware configuration. First is the suggestion to perform a more detailed investigation on the effect of LED brightness on the quality of measured PPGs. In this study, only three brightnesses were tested, which means that it is likely that the most optimal brightness has not yet been discovered. Secondly, some care should be taken to investigate the influence of the arrangement of the LEDs on the device on the quality of the PPG data. Although different numbers of LEDs were tested, the diagonal nature in which the LEDs were arranged could potentially influence the quality. Finally, it is suggested that a thorough pressure analysis be conducted using a loadcell to properly quantify the most optimal sensor-to-skin contact pressure for PPG measurements.

## Figures and Tables

**Figure 1 sensors-25-01875-f001:**
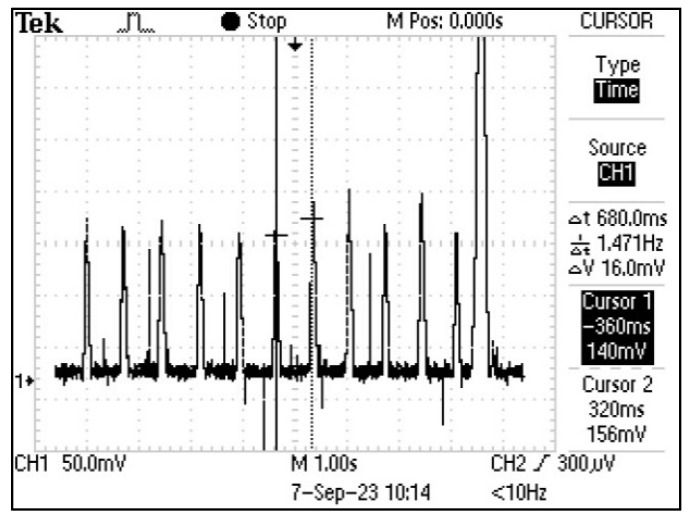
Input circuit output signal of transmissive fingertip PPG measurement in the time domain.

**Figure 2 sensors-25-01875-f002:**
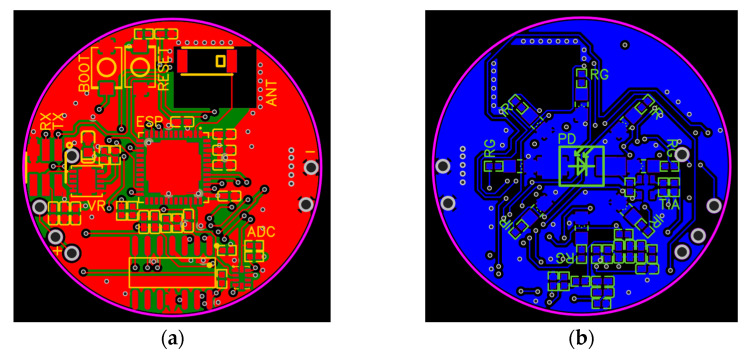
Final PCB design schematic: (**a**) top; (**b**) bottom.

**Figure 3 sensors-25-01875-f003:**
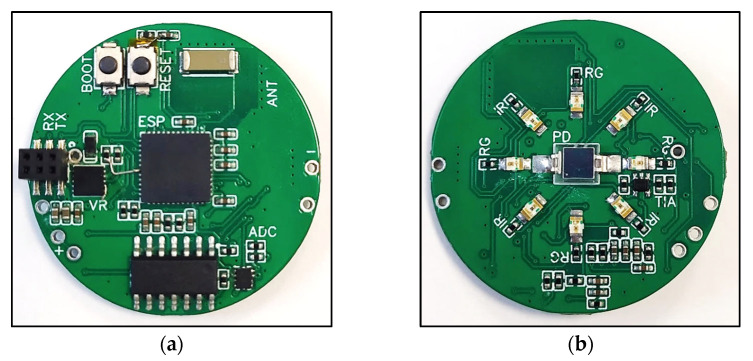
Final PCB design: (**a**) top; (**b**) bottom.

**Figure 4 sensors-25-01875-f004:**
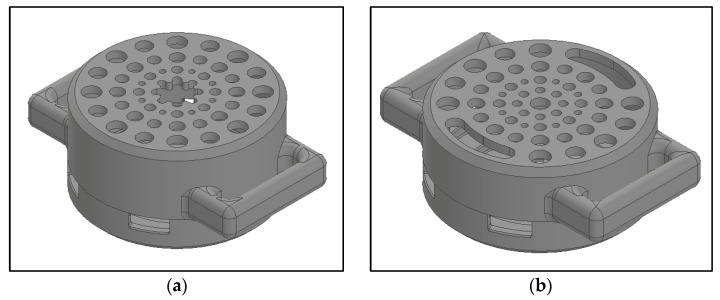
Enclosure design for (**a**) wireless and (**b**) wired applications.

**Figure 5 sensors-25-01875-f005:**
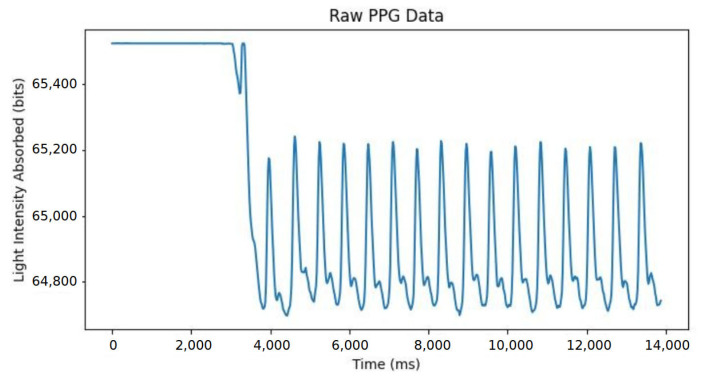
Transient instability example of PPG signal.

**Figure 14 sensors-25-01875-f014:**
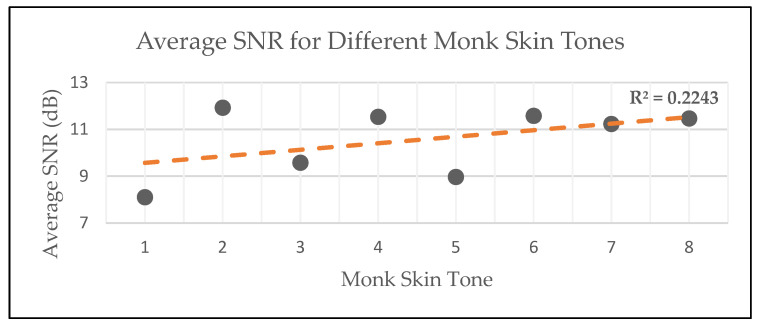
Graphical depiction of average SNR results for different skin tones.

**Table 1 sensors-25-01875-t001:** Resulting configurations of preliminary performance evaluation.

Category	Device 1	Device 2
Body Location	Index Fingertip	Index Fingertip
Light Source Wavelength	Red and Infrared	Green and Infrared
Number of Light Sources	Two Pairs for Red One Pair for Infrared	Two Pairs Green One Pair for Infrared
Light Source Brightness (Duty Cycle)	175 for Red 250 for Infrared	100 for Green 250 for Infrared
Photodetector Lens Shape	Flat	Flat
Sensor-to-Skin Contact Pressure	High	High

**Table 2 sensors-25-01875-t002:** Pearson correlation coefficient of different PPG device configurations (infrared as reference).

Device Configuration	Pearson Correlation Coefficient
Red LED PPG Device	0.846
Green LED PPG Device	0.558

**Table 3 sensors-25-01875-t003:** Average SNR of PPG device configurations (infrared as reference).

Device Configuration	Average SNR (dB)
Red LED PPG Device	10.723
Green LED PPG Device	7.165

**Table 4 sensors-25-01875-t004:** Average SNR results for different skin tones (infrared as reference).

Participants’ Monk Skin Tone	Number of Participants	Average SNR (dB)
1	7	8.11
2	23	11.92
3	19	9.57
4	5	11.53
5	13	8.97
6	19	11.58
7	16	11.23
8	8	11.46

## Data Availability

The original data presented in the participant study, as well as the PCB Gerber and enclosure STL files, are openly available on GitHub at https://github.com/FridgeMagnetPoet/Investigating-the-Design-of-a-Photoplethysmography-Device-for-Clinical-Applications.git (accessed 13 March 2025). Additionally, the complete dataset, including raw and processed data from 110 participants (participant 056 excluded), is openly available in the Mendeley Data repository at Appel, Anneri; Theart, Rensu (2025), “Photoplethysmography Device Data: Hardware and Participant Study Insights”, Mendeley Data, V1, doi: 10.17632/bcb9329f6d.1.

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
