# Peer review of "Investigating the Design of a Photoplethysmography Device for Vital Sign Monitoring"

_sensors, 2025, doi:10.3390/s25061875_

Round 1

Reviewer 1 Report

Comments and Suggestions for Authors

This article is investigating the design of photoplethysmography device for vital sign monitoring. This article was written more likely as a dissertation. The authors should revise this article as a journal paper form than resubmit it again.

Other parts I am concerned about are listed below.

  1. Line 119-133 should be removed to line 100. The introduction should be trimmed in a concise manner.
  2. Line 120 – “Based on Design of Multi- Wavelength Optical Sensor Module for Depth-Dependent Photoplethysmography by Han et al.,” should be quoted correctly, not in this form. Line 126-133 also needs to be corrected.
  3. From figure 1 Transimpedance amplifier design to figure 2 and 3, low pass filter design and high pass filter respectively, these circuits used in this article are very popular and basic for electronic engineering. For a journal paper, authors should emphasize the circuits they used are novel, if not, at least the comparison should be mentioned. For example, as shown in figure 6 and 7, there are four light sources (LEDs) around the detector, what’s the difference between this circuit and other circuits?
  4. Line 236-239, three different wavelengths LEDs were used in this study, however, the penetration depth of these LEDs are different, this factor should be considered and clarified clearly.
  5. Too many descriptions in 3 Software Design, for a journal paper, authors just need to emphasize the contribution they made. What is the novel and new in this paper. Quotation should be one or two sentences.
  6. Line 163-164, The output current of the photodetector is rated between 0 and 37 μA, a 100 Ω resistor was used in circuit as shown in figure 1. The maximum output voltage of OPA170 should be 3.7mV. This voltage is too small to be used by the ADC. This is a mistake.
  1. In 4.9.3. Light Source Wavelength, “the power of red vs green was 0.88, infrared vs green was 1, and infrared vs red was 1.” how to define the ratio? Did the authors measure the output power of each different wavelength LED with a power meter?
  2. In 7 Conclusion, it needs to be trimmed shorter and concise.

Author Response

Reviewer’s Comment 1:

Line 119-133 should be removed to line 100. The introduction should be trimmed in a concise manner.

Author’s Response 1:

The introduction has been significantly streamlined, focusing solely on the essential information relevant to the understanding of the article.

Reviewer’s Comment 2:

Line 120 – “Based on Design of Multi- Wavelength Optical Sensor Module for Depth-Dependent Photoplethysmography by Han et al.,” should be quoted correctly, not in this form. Line 126-133 also needs to be corrected.

Author’s Response 2:

We have corrected the citation and restructured the reference accordingly.

Reviewer’s Comment 3:

From figure 1 Transimpedance amplifier design to figure 2 and 3, low pass filter design and high pass filter respectively, these circuits used in this article are very popular and basic for electronic engineering. For a journal paper, authors should emphasize the circuits they used are novel, if not, at least the comparison should be mentioned. For example, as shown in figure 6 and 7, there are four light sources (LEDs) around the detector, what’s the difference between this circuit and other circuits?

Author’s Response 3:

The figures have been removed as they are also represented in the supplementary material’s circuit design schematics. While the circuit design is not novel, the aim was to share the challenges encountered and insights gained from practical implementation, particularly in terms of the performance differences between theoretical and practical models. Section 2.1 includes a comparison between these models and explains the differences in the final design.

Reviewer’s Comment 4:

Line 236-239, three different wavelengths LEDs were used in this study, however, the penetration depth of these LEDs are different, this factor should be considered and clarified clearly.

Author’s Response 4:

We have added a reference to Section 4.1.1, where the penetration depth and its influence on the PPG signals are clearly explained.

Reviewer’s Comment 5:

Too many descriptions in 3 Software Design, for a journal paper, authors just need to emphasize the contribution they made. What is the novel and new in this paper. Quotation should be one or two sentences.

Author’s Response 5:

As the primary focus of the paper have now shifted to the hardware configuration optimization, most software design details have been removed to ensure the paper emphasizes the novel contributions of our work.

Reviewer’s Comment 6:

Line 163-164, The output current of the photodetector is rated between 0 and 37 μA, a 100 Ω resistor was used in circuit as shown in figure 1. The maximum output voltage of OPA170 should be 3.7mV. This voltage is too small to be used by the ADC. This is a mistake.

Author’s Response 6:

This is not a mistake. The transimpedance amplifier (TIA) does not directly feed into the ADC; rather, its amplification was intentionally reduced due to performance considerations. As detailed in the manuscript, a subsequent low-pass filter (LPF) stage provides additional amplification, ensuring that the final signal level is appropriate before reaching the ADC.

Additionally, it is important to clarify that each circuit component was not individually built and tested with an ADC. Instead, measurements were conducted using an oscilloscope to verify signal behaviour at various stages. The full signal chain, as presented in the circuit schematics, accounts for the necessary amplification before digitization. We attempted to make this point clearer in the text to avoid further confusion.

Reviewer’s Comment 7:

In 4.9.3. Light Source Wavelength, “the power of red vs green was 0.88, infrared vs green was 1, and infrared vs red was 1.” how to define the ratio? Did the authors measure the output power of each different wavelength LED with a power meter?

Author’s Response 7:

The ratios refer to statistical power, not electrical power. The wording has been revised to clarify this distinction.

Reviewer’s Comment 8:

In 7 Conclusion, it needs to be trimmed shorter and concise.

Author’s Response 8:

The conclusion has been revised to make it more concise, in line with your suggestion.

Reviewer 2 Report

Comments and Suggestions for Authors

The experiments appear valuable. However, the paper contains too much content, making it very difficult to read.
It seems that the paper was written before the research was sufficiently completed.

Specific comments:

1. The research objective is unclear as to whether it focuses on "PPG hardware optimization," "performance evaluation of heart rate, oxygen saturation, and blood pressure measurement," or "development of a deep learning model for blood pressure estimation."
-> The inclusion of too many experiments risks dispersing the focus of the paper.

2. The MAE of the blood pressure estimation model (11.28 mmHg, 6.80 mmHg) is relatively high, making it difficult to consider this result as a key contribution of the paper.

3. It may be appropriate to separate the PPG hardware optimization research and the deep learning-based blood pressure estimation research into separate papers.

4. The optimal PPG hardware configuration derived from the preliminary performance evaluation should be directly applied in the main experiment, and excessively detailed hardware experiments should be reduced.

5. The improvements made in the PPG study compared to previous research should be described more specifically.
-> Performance comparisons with other studies are necessary.

6. The CNN-LSTM architecture used for blood pressure estimation needs to be discussed and evaluated more extensively.

7. Consider whether Section 2 on hardware design is necessary in the paper. The excessive length of the paper makes it difficult to read.

8. Section 3 on software design is too unfocused and difficult to read. Unnecessary content should be minimized.

9. Section 4 has a complex structure, making it difficult to read.
->4.7. The evaluation types are too complicated, making it difficult to follow.

10. Section 4 contains a separate conclusion.

11. The content of Section 4 should be condensed for readability, and Section 5 should be emphasized to improve readability.

Author Response

Reviewer’s Comment 1:

The research objective is unclear as to whether it focuses on "PPG hardware optimization," "performance evaluation of heart rate, oxygen saturation, and blood pressure measurement," or "development of a deep learning model for blood pressure estimation."

-> The inclusion of too many experiments risks dispersing the focus of the paper.

Author’s Response 1:

We agree with the reviewer’s observation. The focus of the article has been shifted to PPG hardware optimization, with only brief mentions of the neural network and other aspects to maintain clarity and consistency.

Reviewer’s Comment 2:

The MAE of the blood pressure estimation model (11.28 mmHg, 6.80 mmHg) is relatively high, making it difficult to consider this result as a key contribution of the paper.

Author’s Response 2:

In response to this comment, the focus of the paper has been shifted away from the metric estimations, allowing for a clearer emphasis on the hardware optimization.

Reviewer’s Comment 3:

It may be appropriate to separate the PPG hardware optimization research and the deep learning-based blood pressure estimation research into separate papers.

Author’s Response 3:

We agree with the suggestion and have shifted the focus to hardware optimization in this paper, while deep learning-based blood pressure estimation will be considered for future work, as we have not yet been able to improve on the state of the art.

Reviewer’s Comment 4:

The optimal PPG hardware configuration derived from the preliminary performance evaluation should be directly applied in the main experiment, and excessively detailed hardware experiments should be reduced.

Author’s Response 4:

As the paper now focuses on hardware optimization, the hardware experiments remain included to provide a comprehensive view of the design process and validation, but we have ensured they are relevant to the main objective.

Reviewer’s Comment 5:

The improvements made in the PPG study compared to previous research should be described more specifically.

-> Performance comparisons with other studies are necessary.

Author’s Response 5:

This research is not focused on directly outperforming previous studies but aims to provide a comprehensive exploration of the full PPG hardware system optimization. While we acknowledge the importance of comparisons, the focus has been on documenting the challenges and findings related to hardware optimization, an area with limited documentation in the literature.

Reviewer’s Comment 6:

The CNN-LSTM architecture used for blood pressure estimation needs to be discussed and evaluated more extensively.

Author’s Response 6:

As the focus of the paper has been shifted to hardware design optimization, detailed descriptions of the neural network have been removed to maintain focus on the core topic.

Reviewer’s Comment 7:

Consider whether Section 2 on hardware design is necessary in the paper. The excessive length of the paper makes it difficult to read.

Author’s Response 7:

After careful consideration, we decided that Section 2 on hardware design was essential to the understanding of the research, especially since the primary goal of this work is hardware optimization. As such, it remains in the manuscript, though we have worked to make it more concise.

Reviewer’s Comment 8:

Section 3 on software design is too unfocused and difficult to read. Unnecessary content should be minimized.

Author’s Response 8:

As mentioned earlier, the software design section has been significantly reduced to ensure the paper remains focused on hardware optimization, in line with the revised scope.

Reviewer’s Comment 9:

Section 4 has a complex structure, making it difficult to read.

-> 4.7. The evaluation types are too complicated, making it difficult to follow.

Author’s Response 9:

Thank you for your feedback. Section 4 has been summarized and reorganized to improve clarity and readability, with particular attention given to simplifying the evaluation types.

Reviewer’s Comment 10:

Section 4 contains a separate conclusion.

Author’s Response 10:

We have reworded this section to “Evaluation Conclusion” to emphasize its importance in understanding the hardware optimization process, ensuring it aligns with the overall aim of the paper.

Reviewer’s Comment 11:

The content of Section 4 should be condensed for readability, and Section 5 should be emphasized to improve readability.

Author’s Response 11:

Section 4 has been condensed to focus more closely on the hardware optimization. Given the shift in the focus of the paper, Section 4 now provides more insight into the design process than Section 5, so we have maintained the focus on Section 4.

Reviewer 3 Report

Comments and Suggestions for Authors

The study by Anneri Appel and Rensu P. Theart presented the Design of a Photoplethysmography Device for Vital Sign Monitoring. Manuscript is interesting but the paper has some problems.

Some specific comments:

-The introduction part is too long and needs to be specifics.

-I did not see the purpose of this paper. Methodologically in Introduction section there is no article purpose. The Authors should note the subject of their research and expand this into the purpose of the paper, which is not there yet.

- Results description are presented as a project report, but not as a scientific article. Please correct it.

- The material needs to be shortened and the information needs to be made more specific.

- Authors all times wrote “In this project, this problem was addressed by investigating the”, “this project aims to design” etc. but it is an article/manuscript, please correct it.

- Abstract needs to be specifics.

- Authors wrote “Melanin plays a key role in light absorption in the epidermis. It has been well documented that the absorption of light by melanin increases exponentially towards shorter wavelengths [48], which consequently affects the accuracy of PPG measurements of individuals with darker skin tones” – please explain results for this point. In your research only 6 participants - is it enough for statistical results?

- What about the results if a person has light skin but is tanned?

 - What about the temperature during measurements (exact values)?

- How to take skin tone into account in a hospital setting (real-life scenario)?

- What about results in case of finger deformation?

- Comparison of the performance of the developed Device in terms of limitations and time over existing procedures (in a table) are welcome.

- Some Refs, Accessed 22 July 2024, please correct it for 2025.

- Figs. 12-19 – are poor quality. Improve them. Edit the text part of the figures, increase the font size.

Author Response

Reviewer’s Comment 1:

The introduction part is too long and needs to be more specific.

Author’s Response 1:

The introduction has been significantly trimmed, focusing solely on the information directly relevant to the understanding of the article.

Reviewer’s Comment 2:

I did not see the purpose of this paper. Methodologically in the Introduction section, there is no clear purpose stated. The authors should note the subject of their research and expand on this into the purpose of the paper, which is currently missing.

Author’s Response 2:

This has been addressed by shifting the focus of the article to one specific aspect of the research: hardware optimization. Sections regarding the software, neural networks, and other experiments have either been removed or significantly trimmed.

Reviewer’s Comment 3:

Results description is presented as a project report, not as a scientific article. Please correct it.

Author’s Response 3:

The results section has been revised to focus more on the conclusions drawn from the research, in line with scientific writing conventions.

Reviewer’s Comment 4:

The material needs to be shortened, and the information should be made more specific.

Author’s Response 4:

The manuscript has been trimmed throughout to ensure the content is concise and focused on the key findings.

Reviewer’s Comment 5:

The authors repeatedly write phrases like “In this project, this problem was addressed by investigating…” and “this project aims to design…” but it’s an article/manuscript, so please correct these references.

Author’s Response 5:

The wording has been corrected to reflect the formal tone appropriate for a manuscript, and references to "this project" have been revised.

Reviewer’s Comment 6:

The abstract needs to be more specific.

Author’s Response 6:

The abstract has been revised to be more concise and specific, focusing on the main contributions and findings of the study.

Reviewer’s Comment 7:

The authors wrote, “Melanin plays a key role in light absorption in the epidermis. It has been well documented that the absorption of light by melanin increases exponentially towards shorter wavelengths [48], which consequently affects the accuracy of PPG measurements of individuals with darker skin tones.” Please explain the results for this point. In your research, only 6 participants were included—is that enough for statistical results?

Author’s Response 7:

This has been addressed in Section 4.8.8 (previously Section 4.9.8), where we acknowledge that the sample size is too small to draw statistically significant conclusions regarding skin tone bias. The statement about melanin’s impact on PPG accuracy is based on well-documented literature, but our results are not conclusive due to the limited sample size. This study served as a preliminary validation to ensure the device could measure PPG signals across different participants before conducting a larger-scale study. Future work could expand the sample size to enable statistically significant analysis of this effect.

Reviewer’s Comment 8:

What about the results if a person has light skin but is tanned?

Author’s Response 8:

The skin tone was determined from the inside of the participant’s forearm, which is closely related to the skin tone of the wrist and fingertip. Although a tan does indicate increased melanin at that time, it is not necessarily the participant's normal skin tone. Nevertheless, the results from participants with tanned skin provide valuable information on how melanin affects PPG measurements.

Reviewer’s Comment 9:

What about the temperature during measurements (exact values)?

Author’s Response 9:

All measurements were conducted in a temperature-controlled room, maintained at 21-23°C. As a result, ambient temperature was not considered a variable in this study and was not measured directly. While variations in skin temperature could theoretically influence the electronics, the component datasheets suggest that this effect is minimal within the controlled environment used. However, we acknowledge this as a potential factor for further investigation in future studies

Reviewer’s Comment 10:

How would skin tone be taken into account in a hospital setting (real-life scenario)?

Author’s Response 10:

Our goal is to develop hardware and algorithms that achieve accurate PPG measurements regardless of skin tone. Our study included an analysis of robustness across different participants, and in a hospital setting, the controlled environment and reduced patient movement would likely enhance this robustness.

We acknowledge that previous studies have reported lower PPG accuracy for individuals with darker skin tones. However, our results did not indicate such an effect, potentially due to differences in hardware design, signal processing techniques, or improved algorithms. Further studies with a larger sample size are needed to confirm this.

In real-world hospital applications, ensuring accurate readings across skin tones may involve using multi-wavelength PPG sensors, adaptive algorithms that compensate for skin tone variations, or calibration protocols for different patient groups. These factors would need to be considered in clinical deployment. This was, however, not the focus of this study.

Reviewer’s Comment 11:

What about results in the case of finger deformation?

Author’s Response 11:

This issue was beyond the scope of the current study, as the participant pool was based on voluntary participation and did not specifically include individuals with finger deformities. However, as long as there is sufficient blood flow in the finger our device would still be robust in that scenario. This is, however, an important consideration for future work.

Reviewer’s Comment 12:

A comparison of the performance of the developed device in terms of limitations and time versus existing procedures (in a table) would be welcome.

Author’s Response 12:

A direct performance comparison with commercially available devices was not conducted, primarily because most commercial devices do not provide raw PPG data but rather pre-processed outputs, such as heart rate and blood oxygen saturation. These processed values are heavily influenced by proprietary software algorithms, which are beyond the scope of this study.

Since our focus was on hardware development, we employed only post-processing analysis of the collected data, which was compared to supplementary measurements from a commercial pulse oximeter and blood pressure cuff. Performance in terms of time is inherently a software-dependent factor, and other limitations—such as robustness to varying contact pressures and stray light rejection—are difficult to compare without large-scale data collection. While these factors are discussed in this study, a direct, one-to-one comparison is not feasible under the current scope.

Future work could involve a more structured benchmarking study once full data extraction and processing capabilities are implemented.

Reviewer’s Comment 13:

Some references list “Accessed 22 July 2024,” but this should be corrected for 2025.

Author’s Response 13:

As the research was conducted between 2023 and 2024, the references list the dates when the sources were actually accessed during that time. The access date reflects when the data was retrieved, not when the paper is submitted. Therefore, updating them to 2025 would be incorrect unless the sources were revisited and verified in 2025.

Reviewer’s Comment 14:

Figs. 12-19 are of poor quality. Please improve them and increase the font size of the text in the figures.

Author’s Response 14:

The figures have been revised and improved in quality, and the font size has been increased for better readability.

Reviewer 4 Report

Comments and Suggestions for Authors

This article presents the design, development, and validation process of a Photoplethysmography (PPG) device. The device is intended for use in wearable technology to measure heart rate, blood oxygen saturation, and blood pressure. The study is clearly presented, the results are reliable, and the findings have significant practical implications, contributing to cardiovascular monitoring in populations. After addressing some minor shortcomings, the paper can be accepted for publication.

1. The narrative between lines 101 and 118 in the introduction is somewhat redundant and cumbersome. I suggest the author simplify this section. Similarly, the passage between lines 119 and 133 also needs to be streamlined. Additionally, when citing references, the title of the articles should be avoided.

2. The content on circuit design in section 2.1 needs to be streamlined. The excessive details may hinder the reader's understanding of the overall design approach. The author could move some of the details to the supplementary materials. Figure 5 and its discussion could also be moved to the supplementary materials.

3. In section 3.4, the author should provide a diagram of the neural network structure, labeling the inputs and outputs, to help readers better understand the neural network.

4. Since section 4.1 does not involve any results, I suggest the author simplify this part of the narrative.

5. The results section should focus only on the key conclusions, rather than the research process. The author needs to reduce the content in the conclusion part.

Author Response

Reviewer’s Comment 1:

The narrative between lines 101 and 118 in the introduction is somewhat redundant and cumbersome. I suggest the author simplify this section. Similarly, the passage between lines 119 and 133 also needs to be streamlined. Additionally, when citing references, the title of the articles should be avoided.

Author’s Response 1:

The introduction has been significantly trimmed, focusing only on information relevant to the understanding of the article. The issue with citing article titles has also been addressed and corrected.

Reviewer’s Comment 2:

The content on circuit design in section 2.1 needs to be streamlined. The excessive details may hinder the reader's understanding of the overall design approach. The author could move some of the details to the supplementary materials. Figure 5 and its discussion could also be moved to the supplementary materials.

Author’s Response 2:

As suggested, the figures related to the circuit design have been removed, as they are already included in the supplementary materials. Additionally, the pole-zero analysis has been moved entirely to the Appendix for clarity and brevity.

Reviewer’s Comment 3:

In section 3.4, the author should provide a diagram of the neural network structure, labeling the inputs and outputs, to help readers better understand the neural network.

Author’s Response 3:

As the focus of the article has shifted to hardware design optimization, detailed descriptions and diagrams of the neural network have been removed from the manuscript.

Reviewer’s Comment 4:

Since section 4.1 does not involve any results, I suggest the author simplify this part of the narrative.

Author’s Response 4:

Section 4.1 has been significantly trimmed by removing unnecessary background information, ensuring it is more concise and relevant to the overall narrative.

Reviewer’s Comment 5:

The results section should focus only on the key conclusions, rather than the research process. The author needs to reduce the content in the conclusion part.

Author’s Response 5:

The results section has been revised to focus more on the key conclusions, and unnecessary details about the research process have been reduced.

Round 2

Reviewer 1 Report

Comments and Suggestions for Authors

This revised article has been matched the suggestions I mentioned. 

It can be accepted in the current version. 

Author Response

No changes were requested. Thank you for your valuable feedback.

Reviewer 2 Report

Comments and Suggestions for Authors

Although the overall length of the paper has been significantly reduced, the results should have been concluded in Section 4.8. Instead, Section 5 begins again, and additional results are presented in Section 6. 
To conform to the paper’s formatting, the results from Section 4.8 and Section 6 should be consolidated.

Given that the issues I pointed out have not been addressed, this poses a serious problem. Furthermore, 
after the author responds to these issues, a new revision should be started for re-evaluation.

Author Response

Reviewer’s Comment 1:

Although the overall length of the paper has been significantly reduced, the results should have been concluded in Section 4.8. Instead, Section 5 begins again, and additional results are presented in Section 6.

To conform to the paper’s formatting, the results from Section 4.8 and Section 6 should be consolidated.

Author’s Response 1:

Thank you for your feedback. To improve the paper’s flow and clarity, we have refined Section 4 to emphasize its role in shaping the participant study rather than presenting final results (also note that Section 4.8 is now Section 4.4). These preliminary insights were instrumental in narrowing down the hardware configurations for testing and were a necessary step in our methodology.

To further align with the paper’s formatting, we have also moved some methodological details of Section 4 to Appendix B, ensuring that Section 4 is not perceived as reporting on final results but rather provides context for the participant study. Additionally, we have consolidated the previous Sections 5 and 6 into a single Section 5, streamlining the presentation of our findings while maintaining logical coherence. We believe this restructuring improves readability and ensures a clearer distinction between preliminary insights and final results.

Reviewer’s Comment 2:

Given that the issues I pointed out have not been addressed, this poses a serious problem. Furthermore, after the author responds to these issues, a new revision should be started for re-evaluation.

Author’s Response 2:

We sincerely apologize for not fully addressing the issues raised in the previous round of reviews. In this revision, we have carefully reconsidered your feedback from the first revision and made further substantial changes to ensure a more focused and coherent manuscript.

In addition to the structural refinements outlined in Response 1—where we streamlined Sections 4 and 5 for clarity and improved the distinction between preliminary insights and final results—we have now fully aligned the paper’s scope with PPG hardware optimization and its performance across different skin tones. To achieve this:

  1. Complete Removal of Neural Network Discussions: We have eliminated all content related to neural networks, including the CNN-LSTM model and blood pressure estimation, as these aspects were not central to the core focus of PPG hardware optimization. This ensures that the manuscript is now entirely dedicated to the study’s primary objective.
  2. Further Enhancements to Readability: Beyond merging Sections 5 and 6, we have revisited the entire manuscript to refine the presentation of findings and ensure a clear and logical progression. We have also reduced unnecessary complexity in Section 4 and relocated supplementary methodological details to Appendix B.

We appreciate your patience and constructive feedback, which has been instrumental in improving the clarity and focus of our work. We are confident that this revised version directly addresses the concerns raised and look forward to your re-evaluation.

Reviewer 3 Report

Comments and Suggestions for Authors

-The Results parts needs to be specifics and logically structured. Some section can excluded from the main text of the article and included in the Supplementary Materials section, for ex., 4.8.4. Light Source Brightness, 4.8.5. Photodetector Lens Shape, 4.8.6. Sensor-to-Skin Contact Pressure, 4.8.7.Skin Tone, 5.1. Experimental Conditions, Reference Devices and 5.4. Methodology.

-The Conclusion section should be re-restructured. Specifically, not need to repeat the content of the article. It is necessary to make a conclusion on the article.

-For demonstrate the novelty and originality of the research, please add the comparison of the performance of the developed device with another systems (not only commercial systems).

- I did not see the Supplementary Materials and Figures S. Please add the file with Supplementary Materials.

-Please give information about the statistical method. Describing of the error calculation in Experiments Section is missing.

-Since the device is reusable, how to treat (treatment after each measurements before use on another patient) this device after use without result quality change?

- Some references list “Accessed 22 July 2024,” but this should be corrected for 2024. Authors Response “As the research was conducted between 2023 and 2024, the references list the dates when the sources were actually accessed during that time. The access date reflects when the data was retrieved, not when the paper is submitted. Therefore, updating them to 2025 would be incorrect unless the sources were revisited and verified in 2025” -  but some Refs is valid in 2025. Please correct it for Readers.

Author Response

Reviewer’s Comment 1:

The Results parts needs to be specifics and logically structured. Some section can excluded from the main text of the article and included in the Supplementary Materials section, for ex., 4.8.4. Light Source Brightness, 4.8.5. Photodetector Lens Shape, 4.8.6. Sensor-to-Skin Contact Pressure, 4.8.7.Skin Tone, 5.1. Experimental Conditions, Reference Devices and 5.4. Methodology.

Author’s Response 1:

Thank you for your valuable suggestion. To improve the logical structure of the results section and enhance readability, we have moved the sections on "Experimental Conditions," "Reference Devices," "Methodology," and "Performance Evaluation Metrics" from Sections 4 and 5 to Appendices B and C, respectively. Additionally, we have merged Section 6 with Section 5 to create a more streamlined and coherent presentation of our findings.

However, we have retained the subsections from Section 4.8 (now Section 4.4) that detail insights from the preliminary performance evaluation. These insights were instrumental in shaping the participant study and refining the tested hardware configurations. We believe they provide essential context for understanding the rationale behind the study’s design and are therefore valuable to readers.

We appreciate your feedback and believe these revisions contribute to a clearer and more logically structured manuscript.

Reviewer’s Comment 2:

The Conclusion section should be re-restructured. Specifically, not need to repeat the content of the article. It is necessary to make a conclusion on the article.

Author’s Response 2:

Thank you for your constructive feedback. In response, we have restructured the Conclusion to eliminate redundancy and ensure it provides a concise summary of key findings and their implications rather than reiterating the article's content. The revised Conclusion now highlights the importance of hardware optimization, presents the primary outcomes of the study, and emphasizes the impact of our findings on PPG device design for clinical applications. We believe this revision improves clarity and aligns with the purpose of a conclusion.

Reviewer’s Comment 3:

For demonstrate the novelty and originality of the research, please add the comparison of the performance of the developed device with another systems (not only commercial systems).

Author’s Response 3:

We appreciate your suggestion. While we recognize the potential value of comparing our developed device to other non-commercial (hobbyist) PPG systems, such a comparison was not within the original scope of this study. Our primary objective was to systematically evaluate how different hardware configurations influence the raw PPG signal, particularly across varying skin tones—a level of analysis that most hobbyist devices do not allow.

Unlike our device, most hobbyist PPG systems are designed as fixed-function devices, meaning they do not provide the ability to adjust key parameters such as light source wavelength, photodetector lens shape, or sensor-to-skin contact pressure—all of which were central to our investigation. Furthermore, these devices typically process and filter the PPG signal internally, outputting pre-processed heart rate or SpOâ‚‚ values rather than providing access to the raw PPG signal. This fundamental difference makes a direct comparison difficult, as it would not provide meaningful insights into the core focus of our study: how hardware modifications impact raw PPG signal acquisition.

Additionally, conducting a rigorous comparison would require a separate study in which we retest our developed device alongside various hobbyist systems on a new participant group under controlled conditions. Given that our study has already been completed, including data collection and analysis, such an addition is not feasible at this stage.

However, we acknowledge the importance of benchmarking and encourage future work that explores comparative performance across a broader range of PPG systems, particularly in the context of skin-tone inclusivity and raw signal fidelity—areas that are often overlooked in existing devices.

Reviewer’s Comment 4:

I did not see the Supplementary Materials and Figures S. Please add the file with Supplementary Materials.

Author’s Response 4:

We apologize for any confusion regarding the Supplementary Materials. To our understanding, the Supplementary Material was included during the initial submission (and revision submission), but we recognize that it may not have been easily accessible on your end. To ensure clarity, we have resubmitted the Supplementary Material as a separate file alongside this revision. Please let us know if there are any issues accessing it.

Reviewer’s Comment 5:

Please give information about the statistical method. Describing of the error calculation in Experiments Section is missing.

Author’s Response 5:

Thank you for your feedback. The error calculation methodology is described in the newly renumbered Section 5.1 (PPG Device Configuration), where we explain how Pearson correlation coefficient (PCC) and signal-to-noise ratio (SNR) were used to evaluate signal accuracy and quality.

The PCC quantifies the strength of the relationship between the test signals and the reference infrared PPG signals, while SNR measures the signal strength relative to noise, providing a direct assessment of signal quality. These two metrics inherently capture the accuracy and reliability of the recorded signals.

We appreciate your suggestion and trust that this section adequately details the statistical approach used in our study.

Reviewer’s Comment 6:

Since the device is reusable, how to treat (treatment after each measurements before use on another patient) this device after use without result quality change?

Author’s Response 6:

Thank you for your valuable comment. We overlooked documenting the cleaning procedure in the original manuscript. After each measurement, the device was cleaned using 91% rubbing alcohol to ensure proper hygiene and prevent any compromise in results. This information has now been added to the methodology section for clarity.

Reviewer’s Comment 7:

Some references list “Accessed 22 July 2024,” but this should be corrected for 2024. Authors Response “As the research was conducted between 2023 and 2024, the references list the dates when the sources were actually accessed during that time. The access date reflects when the data was retrieved, not when the paper is submitted. Therefore, updating them to 2025 would be incorrect unless the sources were revisited and verified in 2025” -  but some Refs is valid in 2025. Please correct it for Readers.

Author’s Response 7:

We appreciate your attention to detail. We have updated all internet references to reflect "Accessed 3 March 2025" where applicable, ensuring consistency and accuracy for the readers.

Round 3

Reviewer 2 Report

Comments and Suggestions for Authors

I appreciate the significant effort put into substantially restructuring the paper based on my suggestions. However, the comparisons are primarily qualitative, and since most of Section 4 focuses on these qualitative topics, it is hard to argue that the paper meets the necessary standards. Nevertheless, I believe that with further research, this work could develop into a good paper.

Specific comments:

1. In Figure 6, it appears that the signal from the wrist is considered worse due to more noise, but this is not supported by any quantitative comparison. Since the comparison is qualitative, the reasoning needs to be more logically substantiated for the paper to be acceptable.

2. In Figures 7 and 8, how were the two signals compared? A quantitative, rather than qualitative, comparison is necessary.

3. On line 385, the statement “Qualitative analysis was therefore prioritized to explore nuances that might not be captured quantitatively” indicates that the analysis is based on the author’s subjective judgment.

4. On line 373, the statement “The infrared LED, however, performed better than both the visible-spectrum LEDs on a qualitative basis” shows that the selection was made intuitively, rather than through quantitative comparison, which renders the paper unsuitable for publication.

5. In Section 4, from lines 249–254, there are too many different settings described, all of which appear to have been selected intuitively. This makes the paper seem unsuitable for publication. Additionally, the statistical analysis in Section 4 is difficult to read and lacks logical clarity.

6. For Figures 9, 10, and 11, the logic behind the comparisons between the figures is unclear. What exactly is the t-test comparing? Which metric’s mean is being compared? This is hard to understand.

7. In Section 5, the concept of SNR is understandable. It is acceptable to compare signals based on SNR.

Author Response

Reviewer’s Comment 1 to 4:

In Figure 6, it appears that the signal from the wrist is considered worse due to more noise, but this is not supported by any quantitative comparison. Since the comparison is qualitative, the reasoning needs to be more logically substantiated for the paper to be acceptable.

In Figures 7 and 8, how were the two signals compared? A quantitative, rather than qualitative, comparison is necessary.

On line 385, the statement “Qualitative analysis was therefore prioritized to explore nuances that might not be captured quantitatively” indicates that the analysis is based on the author’s subjective judgment.

On line 373, the statement “The infrared LED, however, performed better than both the visible-spectrum LEDs on a qualitative basis” shows that the selection was made intuitively, rather than through quantitative comparison, which renders the paper unsuitable for publication.

Author’s Response 1 to 4:

Thank you for your insightful comments and valuable feedback. We fully acknowledge the importance of providing robust quantitative comparisons and understand your concerns regarding the qualitative nature of the analysis presented in Figures 6, 7, and 8.

Due to experimental constraints, we did not simultaneously record a reference (gold standard) PPG signal from an alternative, commercially validated device. One reason for this is the fact that two PPG devices cannot measure simultaneously at the same location of the body. Additionally, our statistical tests (ANOVA, F-, and T-tests), intended to quantify differences among variables, did not yield statistically significant results. Consequently, the reliability of a purely quantitative assessment was limited.

To address this limitation, we initially explored simulating an ideal reference PPG signal based on well-established literature benchmarks, specifically waveform characteristics exhibiting clear systolic and diastolic peaks. However, due to substantial physiological variability across the six participants—including differences in heart rate (which is also variable with time), perfusion, and tissue characteristics—generating a universally applicable, physiologically accurate reference signal was impractical and potentially misleading. Nevertheless, literature clearly defines the characteristics expected of an ideal PPG signal, and these characteristics formed the basis for our qualitative assessment criteria.

Accordingly, we adopted a rigorous qualitative approach guided by clearly defined and predominantly objective rankings (Tables B.1 and B.2). To ensure consistency and minimize bias, all qualitative assessments were systematically conducted by a single researcher in one session, with comprehensive comparisons made collectively across all participants. Representative examples across various quality factor values, in both frequency-domain (FFT) and time-domain visualizations, have been added (Figures B.1 and B.2) to clarify our analytical criteria. We believe these clearly illustrate discernible differences in signal quality, supporting our argument that objective differences can indeed be assessed through our qualitative framework.

Regarding the specific comment on line 373, we acknowledge the phrasing may have implied an intuitive selection. To clarify, the infrared LED signal was systematically chosen based on consistent qualitative criteria: enhanced signal clarity, reduced noise artifacts, and overall stability, as thoroughly documented across participants (as detailed above). We have adjusted the manuscript text accordingly to explicitly state that the selection of the infrared LED was based on systematically observed criteria rather than intuition.

We fully recognize the inherent limitations of our qualitative approach and have transparently detailed these constraints within the manuscript. Given the experimental context, we firmly believe the methodological transparency and robust visual evidence provided in Appendix B adequately substantiate our approach. We welcome further recommendations to enhance clarity or address additional concerns.

Reviewer’s Comment 5:

In Section 4, from lines 249–254, there are too many different settings described, all of which appear to have been selected intuitively. This makes the paper seem unsuitable for publication. Additionally, the statistical analysis in Section 4 is difficult to read and lacks logical clarity.

Author’s Response 5:

We thank the reviewer for these valuable observations. Upon careful consideration of your feedback, we have substantially revised Section 4 to address both the concern about hardware settings selection and the clarity of our statistical analysis.

Regarding the selection of hardware settings, we acknowledge that our approach may have appeared intuitive without sufficient justification. We want to clarify that each configuration choice was actually grounded in existing literature (as referenced in Section 1), previous research findings, and practical constraints of available off-the-shelf components. In our revised manuscript, we have made these connections more explicit to demonstrate the scientific rationale behind our design decisions.

With respect to the statistical clarity concerns, we have comprehensively improved the presentation and structure of our statistical analysis throughout Section 4. We added a new introductory paragraph that clearly outlines our statistical methodology, specifying the use of ANOVA and t-tests with a significance level of α = 0.05, and explaining our complementary approach of qualitative analysis when statistical tests were inconclusive. We also restructured the section to more clearly describe our "hierarchical elimination approach" for systematically evaluating different hardware configurations.

Throughout Sections 4.4.2-4.4.6, we have incorporated power analysis information to provide context for our statistical results, particularly in cases where the null hypothesis was not rejected. This addition helps readers understand whether non-significant results were due to insufficient statistical power or genuine lack of differences between configurations. We've also enhanced the interpretation of statistical versus qualitative findings, particularly in Section 4.4.6, where we explain: "The discrepancy between qualitative observations and statistical results suggests that while trends are visually apparent, the variability in the data prevented statistical significance, highlighting the importance of combining both approaches."

Finally, we improved our discussion of sample size limitations in Section 4.4.7 regarding skin tone evaluation to better contextualize our findings and justify our approach to the subsequent participant study. These revisions maintain the necessary brevity while significantly enhancing the scientific rigor and logical clarity of our statistical analysis. We believe these changes address your concerns while preserving the manuscript's focus on the core findings presented in Section 5.

Reviewer’s Comment 6:

For Figures 9, 10, and 11, the logic behind the comparisons between the figures is unclear. What exactly is the t-test comparing? Which metric’s mean is being compared? This is hard to understand.

Author’s Response 6:

Thank you for highlighting this lack of clarity. We acknowledge that our explanation of the statistical methodology related to Figures 9-11 was insufficient.

For these brightness comparison figures, we performed pairwise t-tests comparing the means of our PPG signal quality factors across three brightness levels (duty cycles of 100, 175, and 250) for each LED type (red, green, and infrared). The quality factor was a composite metric derived from both time-domain and frequency-domain characteristics of the PPG signals, where lower values indicate better signal quality with fewer artifacts and clearer physiological features.

Specifically, we conducted three pairwise comparisons for each LED type: 100 vs. 175, 175 vs. 250, and 100 vs. 250. As noted in Section 4.4.4, these tests yielded no statistically significant differences (pANOVA = 0.87584), despite visually observable trends in signal quality that informed our final device configurations. Our power analysis revealed adequate statistical power (>0.9) for most comparisons, with only the 100 vs. 250 comparison being slightly underpowered (0.77).

We have revised the manuscript to explicitly state these details, clarifying both what was being compared and why the statistical results did not align with our qualitative observations. We have also added a reference to Appendix B where readers can find detailed information about the quality factor calculation methodology. These changes should make the logic behind our statistical approach much clearer.

Reviewer 3 Report

Comments and Suggestions for Authors

The authors have made good improvements to the manuscript.

Author Response

(The authors gave the same response as above.)
